# Facile Synthesis of Sandwich-Like rGO/CuS/Polypyrrole Nanoarchitectures for Efficient Electromagnetic Absorption

**DOI:** 10.3390/ma13020446

**Published:** 2020-01-17

**Authors:** Bing Zhang, Shaofeng Lin, Jingjing Zhang, Xiaopeng Li, Xiaodong Sun

**Affiliations:** 1School of Physics, Beijing Institute of Technology, Beijing 100081, China; zb_mat@163.com; 2Information & Communications Institute, National University of Defense Technology, Xi’an 710106, China; linshaofeng316@sina.com (S.L.); destinybit@163.com (J.Z.); asmany@163.com (X.L.); 3Marine Department of Satellite Tracing and Metering, Jiangyin 214400, China

**Keywords:** reduced graphene oxide, interfacial polarization, dielectric loss, effective bandwidth

## Abstract

Currently, electromagnetic pollution management has gained much attention due to the various harmful effects on wildlife and human beings. Electromagnetic absorbers can convert energy from electromagnetic waves into thermal energy. Previous reports have demonstrated that reduced graphene oxide (rGO) makes progress in the electromagnetic absorption (EA) field. But the high value of permittivity of rGO always mismatches the impedance which results in more electromagnetic wave reflection on the surface. In this work, sandwich-like rGO/CuS/polypyrrole (PPy) nanoarchitectures have been synthesized by a facile two-step method. The experimental result has shown that a paraffin composite containing 10 wt.% of rGO/CuS/PPy could achieve an enhanced EA performance both in bandwidth and intensity. The minimum reflection loss (RL) value of −49.11 dB can be reached. Furthermore, the effective bandwidth can cover 4.88 GHz. The result shows that the as-prepared rGO/CuS/PPy nanoarchitectures will be a promising EA material.

## 1. Introduction

The swift expansion of communication devices, such as mobile phone, wire-less internet as well as military equipment, have caused an increase in electromagnetic pollution [1,2,3,4,5,6]. Because of its harmful effects on the environment and organisms, it has become a serious danger for modern society. To solve this problem, electromagnetic absorbers have become indispensable. They are a kind of material that can provide an alternative to the attenuation of electromagnetic waves [7,8,9,10]. For practical applications, qualified electromagnetic absorbers should possess several characteristics such as wide absorption, strong electromagnetic absorption (EA) capability, high thermal stability, antioxidant capacity, and low weight [11,12,13,14,15]. In the design process, strong attenuation capability and proper impedance matching are important factors for high-performance absorbers. Generally, if |*Z*_in_/*Z*_0_| = 1 (*Z*_0_ = 376.7 Ω is the intrinsic impedance of free space, *Z*_in_ is the input characteristic impedance of the absorber), an electromagnetic wave can hardly reflect on the surfaces of the material [16,17,18,19].

Carbon materials are representative absorbers, beneficial for their dielectric loss, but the high value of their conductivity from this kind material always makes their impedance mismatch [20,21,22]. According to a previous report [23], the calculated minimum reflection loss (RL) of graphene can hardly reach up to −10 dB. As a traditional absorber, ferromagnetic materials have been used since the First World War. However, practical applications have demonstrated that ferromagnetic materials have a lot of drawbacks such as high density, large disparities in electromagnetic parameters, and poor anticorrosion ability [24,25,26].

Advances in the field of EA studies has increasingly paid attention to features such as light weight, low density, strong absorption intensity, and broad effective bandwidth [27,28]. Numerous multiple-component materials have been reported to absorb electromagnetic waves. For example, Feng et al. [29] synthesized ZnFe_2_O_4_@rGO@TiO_2_ microspheres with a yolk-shell structure. The effective absorption bandwidth can cover 4.1 GHz, and a minimum RL value of −44.3 dB can be achieved at 15.92 GHz. Pan et al. [30] prepared the novel Fe_3_O_4_@SiO_2_@rGO nanocomposite, and a paraffin composite with 20 wt.% Fe_3_O_4_@SiO_2_@rGO showed enhanced EA properties. A minimum RL value of −26.6 dB can be achieved at 9.7 GHz, and the effective absorption bandwidth can cover from 4.4 to 17.3 GHz (2–5 mm). A tunable method for the preparation of multi-interfacial materials is desired in the field of EA, because the composite can produce extensive interfacial polarization to absorb electromagnetic waves, and such property is not available in the single-phase.

Graphene has already obtained an enormous reputation because of its wide range of practical applications. Recently, graphene has been at the forefront in the field of materials research and is remarkable due to the fact of its high carrier mobility, high surface area, low density, and abundant defects [31,32,33,34,35,36]. Generally, graphene is called reduced graphite oxide (rGO). In the EA area, it has always been utilized as a substrate to synthesize rGO-based absorbers. In recent years, numerous studies have focused on the combination of rGO with magnetic materials to construct EA materials. However, in the frequency of 2–18 GHz, large-scale magnetic materials always exhibit low permeability with little fluctuation which is difficult to handle while the permittivity can obtain modulation in a simple way. Thus, it is reasonable to fabricate EA materials by modulating the dielectric’s permittivity. Summarily, it is a permittivity regulating strategy. Quan et al. [23] prepared MoS_2_/rGO composites using an effective hydrothermal method. The minimum RL value of −67.1 dB could be obtained at 14.8 GHz, and the effective EA bandwidth could cover 5.92 GHz (12.08–18.00 GHz). The results fully demonstrated that Quan’s [23] work acquired absorber with strong attenuation ability and suitable impedance matching simultaneously. Feng et al. [37] constructed an rGO/ZnO composite through an in situ crystallization approach. The composite showed an enhanced EA performance with high absorption intensity (minimum RL value of −77.5 dB) and broad effective EA bandwidth (6.9 GHz). In light of the existing literature, in this work, we used a facile two-step method and successfully prepared a ternary rGO/CuS/PPy absorber with sandwich-like nanoarchitectures. As far as we know, this ternary composite has rarely been reported to date. We aimed to construct an absorber with the consideration of both moderate impedance matching and attenuation and attenuation capability. In addition, the EA enhancement in the present work can also be originated from the intensified polarization of the rGO–CuS interface, rGO–PPy interface as well as CuS–PPy interface. Based on the results, the as-prepared sandwich-like rGO/CuS/PPy absorber exhibited the features of lightweightedness, strong absorption intensity, and wide effective EA bandwidth which satisfied the requirements of an excellent EA material.

## 2. Materials and Methods

### 2.1. Materials

All chemical regents applied in this study were of analytical grade. The graphene oxide (GO) powder used in this work was provided from XFNANO, XFNANO Materials Tech Co., Ltd., Nanjing, China. Pyrrole (Py) monomer, cupric acetate monohydrate (Cu(CH_3_COO)_2_ H_2_O), and thiourea were purchased from GENERAL-REAGENT. Ethanol, ferric chloride hexahydrate (FeCl_3_ 6H_2_O), and ethylene glycol (EG) were provided from Shanghai Sinopharm Chemical reagent Co. Ltd., Shanghai, China.

### 2.2. Synthesis of Ternary rGO/CuS/PPy

Sandwich-like rGO/CuS/PPy nanoarchitectures were prepared by a two-step approach. In the first step, 35 mg GO and 150 mg Cu(CH_3_COO)_2_ H_2_O were dissolved in EG (70 mL) and ultra-sounded for 2 h to achieve a uniform dispersion. One hundred and seventy milligrams of thiourea were then added to the mixed solution and magnetically stirred for 30 min. After the reaction, the mixture was transferred into a Teflon-lined stainless-steel autoclave. The autoclave was maintained at 180 °C for 12 h and then allowed to cool to room temperature. The resulting composite (rGO/CuS) was washed by ethanol and deionized water until neutral by centrifugal machine. In the second step, 30 mg rGO/CuS and 135 mg pyrrole was ultrasonically dispersed in a solution containing 2 mL ethanol and 2 mL deionized water. Then 1.6 g FeCl_3_ 6H_2_O mixed in 1.0 mL absolute ethanol and 1.0 mL deionized water was added into the rGO/CuS/pyrrole solution. The mixed solution was mechanical stirred for 5 min, and the final product was obtained by aging for 24 h. The rGO/CuS/PPy hybrid was taken out and washed to remove the impurities. At last, the composite was dried at 60 °C in vacuum.

### 2.3. Characterization and Measurement

The Raman spectroscopy was collected on a Renishaw in via Raman Microscope equipped with an excitation line of 532 nm. The structural analysis of the products was collected by X-ray diffraction (XRD, D8-Advance, Bruker, Karlsruhe, Germany) using a Cu Kα radiation (λ = 0.15418 nm) in the scattering range (2θ) of 10–80^°^ at an accelerating voltage of 40 kV. Transmission electron microscopy (TEM, JEM-2100F, JEOL, Tokyo, Japan) and scanning electron microcopy (FE-SEM, S-4800, Hitachi, Tokyo, Japan) were used to characterized the morphologies. In the frequency range of 2–18 GHz, the electromagnetic parameters of permittivity (*ε*_r_ = *ε*’ − *jε*”) and permeability (*μ*_r_ = *μ*’ − *jμ*”) were collected by vector network analyzer (VNA, N5242A PNA-X, Agilent, Agilent Technologies Inc., Santa Clara, CA, USA). The samples prepared in this study were mixed with wax and pressed into toroidal-shaped samples.

Base on the transmission line theory, the following equations can be used to calculate the theoretical RL [38,39]:(1)$$\mathrm{RL}=20\log|\left(Z_{in}-Z_{0}\right)/\left(Z_{in}+Z_{0}\right)|$$
(2)$$Z_{in}=Z_{0}\sqrt{\frac{\mu_{r}}{\varepsilon_{r}}}\tanh\left(j\frac{2\pi fd\sqrt{\mu_{r}\varepsilon_{r}}}{c}\right)$$
where *Z*_0_ and *Z_in_* are the impedance of free space and the normalized input impedance of absorber, *c* is the velocity of light, *d* is the thickness, and *f* is the frequency. In general, a bandwidth lower than −10 dB (means more than 90% of the electromagnetic energy was absorbed) considered as an effective EA bandwidth [40,41].

## 3. Results and Discussion

### 3.1. Characterization of Samples

The XRD spectra of rGO/CuS/PPy is given in Figure 1a. The diffraction peaks at 27.9°, 29.3°, 31.6°, 33.1°, 47.8°, 52.4°, 59.6°, and 74.0° can be attributed to (101), (102), (103), (006), (110), (108), (116), and (208) planes of CuS according to PDF 85-0620. The diffraction peak widened which may be caused by the small grain structure of the sample. It can be observed that a broad peak appeared near 26° and covered the characteristic peak of graphene (002) near 25° which indicated that the PPy in rGO/CuS/PPy had an amorphous structure, and the scattering pair of protic interplanar molecular chains of PPy should have a diffraction peak in this range. In general, chemically synthesized PPy was linked to α–α bonds, but some α–β bonds can cause molecular disorganization and lead to amorphous PPy. There were no other impurity peaks in Figure 1, indicating that the product prepared in the experiment was of high purity. Raman spectral characteristic peaks of carbon materials were D-band at 1341 cm^−1^ and G-band at 1574 cm^−1^. The D peak was caused by defects in the graphite edge and inside the graphite, while the G peak was caused by the in-plane stretching vibration of carbon atom sp^2^ hybridized in the same plane. The strength ratio of peak D to peak G (*I*_D_/*I*_G_) can be used to study the crystallinity of carbon materials, and the higher *I*_D_/*I*_G_ value represents the crystallinity of the lower carbon. Figure 1b shows the Raman spectrum of rGO/CuS/PPy. After calculation, the *I*_D_/*I*_G_ value of the composite was 1.02, higher than the *I*_D_/*I*_G_ value of GO (0.93), indicating that the graphitization degree of rGO/CuS/PPy increased and GO was successfully restored to rGO in this experiment.

Figure 2 shows the SEM diagram of rGO/CuS/PPy. The result shows that the product morphology can be clearly observed in which dense PPy particles evenly cover the surface of graphene and forming a “sandwich” structure. It can be observed from the enlarged SEM that the surface of graphene was rough and some excessive PPy particles condensated together to form an irregular banded structure. No exposed graphene surface and CuS particles growing on the graphene surface were found in the SEM of rGO/CuS/PPy, indicating that PPy completely enveloped the RGO/CuS composite. The TEM was used to further characterize the morphology and dimension of rGO/CuS/PPy ternary composites prepared in this experiment, and the results are shown in Figure 3. As can be seen from the figure, both the graphene and PPy were translucent, and the dark fringes of the graphene sheet corresponded to excessive PPy, and CuS particles loaded on the graphene surface were clearly visible (in the dark shaded area) which is consistent with the results of the SEM. In addition, the distribution diagram of EDX elements (Figure 4) shows the uniform distribution of C, N, Cu, and S elements on the surface of graphene, further proving the formation of rGO/CuS/PPy ternary.

### 3.2. Electromagnetic Absorption Property

To reveal the electromagnetic absorption mechanism of rGO/CuS/PPy, the sample was mixed with paraffin in different proportions and made into a coaxial ring. The electromagnetic parameters of the material were measured by VNA in the frequency range of 2–18 GHz, and the results are shown in Figure 5. Because rGO/CuS/PPy lacks magnetic components, the real part of its complex permeability was close to constant 1, while the imaginary part was close to 0. Samples with different filler loading were tested in the frequency of 2–18 GHz. 5 wt.% of rGO/CuS/PPy ε’ decreased from 6.24 to 3.44, 10 wt.% of rGO/CuS/PPy ε’ decreased from 9.93 to 5.57, 15 wt.% of rGO/CuS/PPy ε’ decreased from 15.46 to 7.90, 20 wt.% of rGO/CuS/PPy ε’ fell to 9.38 from 23.28. The above phenomenon was due to the increasing applied electric field frequency which makes the inherent dipole in rGO/CuS/PPy paraffin composite unable to keep up with the changing frequency of the applied electric field, resulting in dielectric dispersion effect, and the ε’ value presents a downward trend. More of the electron transfer between the main and multicomponent and more in the interface of the ternary material related to interfacial polarization effect.

The RL can reflect the EA performance of materials, and its value can be calculated by electromagnetic parameters at different frequencies. The calculation formulas have been given by Equations (1) and (2). As can be seen from Figure 6, the RL peak of all samples moved to the low frequency with the increase of matching thickness. This phenomenon can be attributed to the interference caused by the reflection of electromagnetic waves between the upper and lower interfaces of the coating. When the filler loading was only 5 wt.%, the maximum RL value of the rGO/CuS/PPy was −10.85 dB, and the corresponding effective absorption band was only 1.8 GHz (9.04–10.84 GHz). The bad EA performance of 5 wt.% rGO/CuS/PPy was consistent with lower dielectric loss. For 10 wt.% rGO/CuS/PPy, the maximum RL value reached −49.11 dB at 7.42 GHz when the matching thickness was 4.0 mm. Meanwhile, when the matching thickness was 2.5 mm, the corresponding effective absorption band could reach 4.88 GHz (10.96–15.84 GHz). When the filler loading increased to 15 wt.%, the maximum RL value was −39.55 dB, and the corresponding matching thickness was 1.5 mm. When the matching thickness was 2.0 mm, the effective absorption band could cover 4.84 GHz (11.32–16.16 GHz). The contour maps paraffin composites containing different loading ratio of rGO/CuS/PPy plotted in Figure 7 can more intuitively reflect the EA performance, and it was obvious that 10 wt.% and 15 wt.% rGO/CuS/PPy had better EA performance. In addition, the amount of rGO/CuS/PPy filler loading in the paraffin matrix was only 10 wt.% and 15 wt.%, which were far less than most of the currently reported EA materials.

Degree of impedance matching can be revealed by impedance matching ratio *Z* (*Z* = *Z_in_*/*Z*_0_). A higher *Z* value implies the preferable EA property, *Z* = 1 indicates the incident wave can enter into the composite entirely with zero-reflection on the surface. In addition, the energy attenuation should also be taken into account. The attenuation constant (*α*) can be expressed as follow [42]:(3)$$\alpha =\frac{\sqrt{2}\pi f}{c}\times \sqrt{\left(\mu^{\prime \prime }\varepsilon^{\prime \prime }-\mu^{\prime }\varepsilon^{\prime }\right)+\sqrt{{\left(\mu^{\prime \prime }\varepsilon^{\prime \prime }-\mu^{\prime }\varepsilon^{\prime }\right)}^{2}+{\left(\mu^{\prime }\varepsilon^{\prime \prime }+\mu^{\prime \prime }\varepsilon^{\prime }\right)}^{2}}}$$

According to the equations respectively, the *α* and *Z* of rGO/CuS/PPy composite were calculated, and the results are plotted in Figure 8. With the increase of the filler loading, the value of *α* showed an increasing trend. While *Z* showed a decreasing trend with the increase of the filler loading. Because of the relatively low complex permittivity, rGO/CuS/PPy of 5 wt.% had improved impedance matching characteristics, but its attenuation capacity was very weak. On the contrary, rGO/CuS/PPy of 20 wt.% had excellent attenuation characteristics while impedance matching characteristics were very poor. The above results are consistent with the analysis of electromagnetic parameters. Furthermore, it can be seen from the Figure 8 that both rGO/CuS/PPy of 10 wt.% and 15 wt.% have the upper *α* value and *Z* value, so electromagnetic waves can enter the material to the maximum extent and dissipate its energy quickly to achieve a better EA performance.

## 4. Conclusions

In this paper, starting from the design of component regulation, CuS nanoparticles were first compounded on the surface of the rGO nanosheet by solvent heat method, and then a layer of PPy was coated on the surface of rGO/Cus by in situ polymerization, forming an rGO/CuS/PPy ternary material with a “sandwich” structure. Them SEM, TEM, and other tests showed that in the ternary materials, dense PPy particles were uniformly covered on the surface of rGO/CuS by in situ polymerization, forming a “sandwich” structure. The rGO/CuS/PPy of 10 wt.% had the maximum RL of −49.11 dB at a matching thickness of 4.0 mm. When the matching thickness was 2.5 mm, the corresponding effective EA band reached 4.88 GHz. While the filler loading increased to 15 wt.%, the maximum RL value was −39.55 dB, and the corresponding matching thickness was 1.5 mm. When the matching thickness was 2.0 mm, the effective EA band reached 4.84 GHz. The introduction of multiple components and the unique structure of rGO/CuS/PPy not only improves the dielectric loss capacity and impedance matching characteristics, but also enhances the interface polarization effect thus achieving a better EA performance.

## Figures and Tables

**Figure 1 materials-13-00446-f001:**
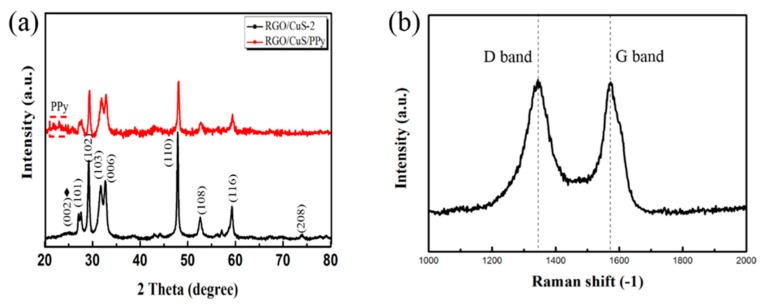
XRD spectra (**a**) and Raman spectra (**b**) of rGO/CuS/PPy sample.

**Figure 2 materials-13-00446-f002:**
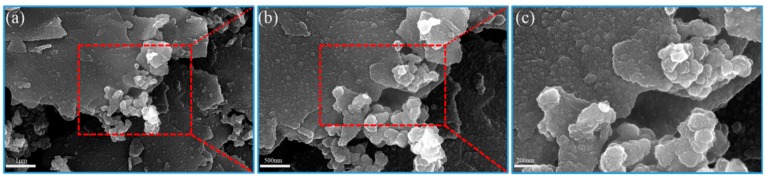
Different magnification SEM images of (**a**–**c**) rGO/CuS/PPy sample.

**Figure 3 materials-13-00446-f003:**
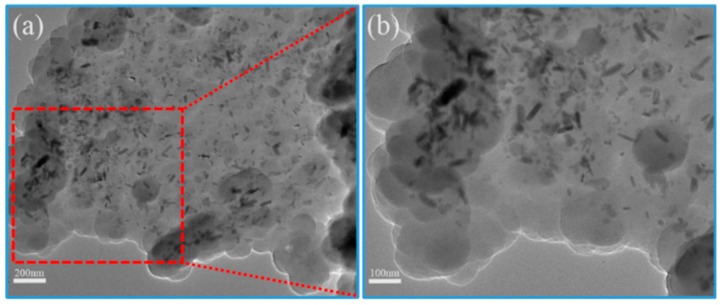
Different magnification TEM images of (**a**,**b**) rGO/CuS/PPy sample.

**Figure 4 materials-13-00446-f004:**
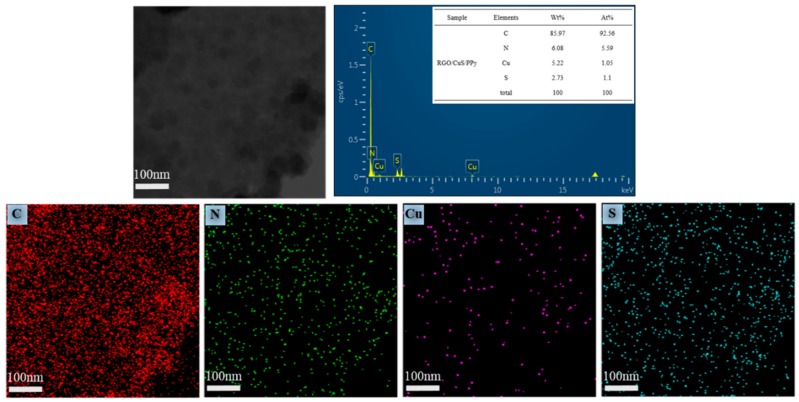
Elemental mappings of C, N, Cu, and S of rGO/CuS/PPy sample.

**Figure 5 materials-13-00446-f005:**
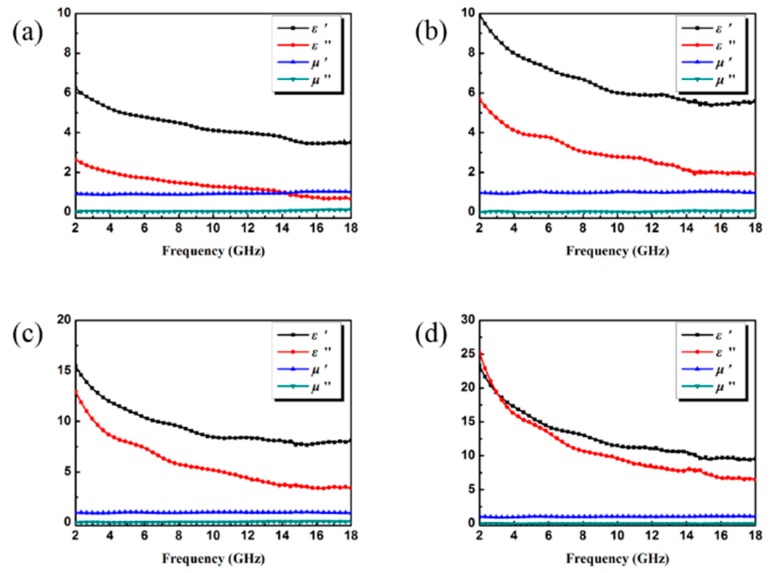
Complex permittivity and permeability of rGO/CuS/PPy with the filler loading of 5 wt.% (**a**), 10 wt.% (**b**), 15 wt.% (**c**), and 20 wt.% (**d**).

**Figure 6 materials-13-00446-f006:**
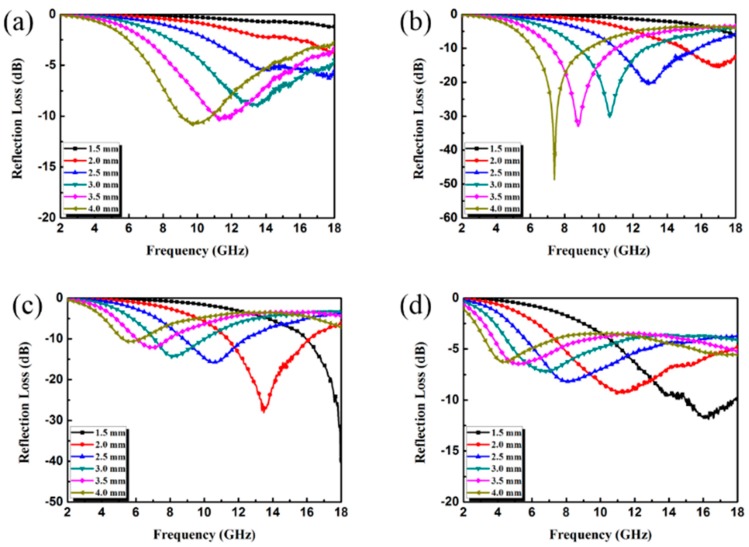
RL curves of paraffin composites containing 5 wt.% (**a**), 10 wt.% (**b**), 15 wt.% (**c**), and 20 wt.% (**d**) of rGO/CuS/PPy.

**Figure 7 materials-13-00446-f007:**
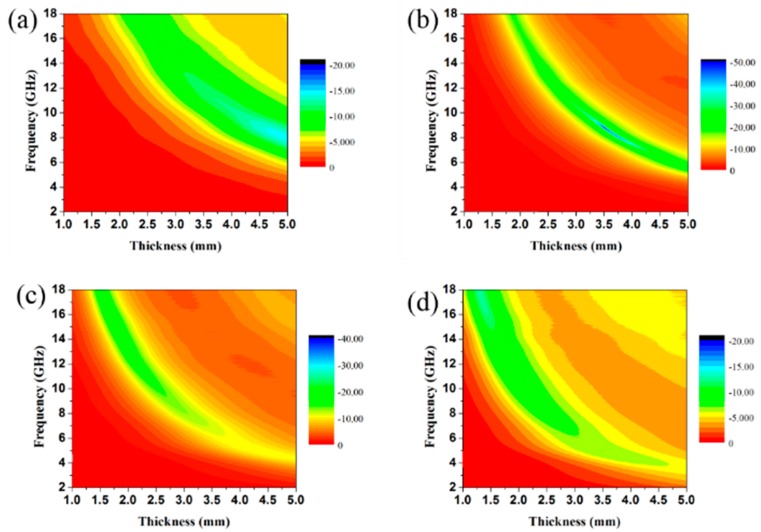
RL contour maps of paraffin composites containing 5 wt.% (**a**), 10 wt.% (**b**), 15 wt.% (**c**), and 20 wt.% (**d**) of rGO/CuS/PPy.

**Figure 8 materials-13-00446-f008:**
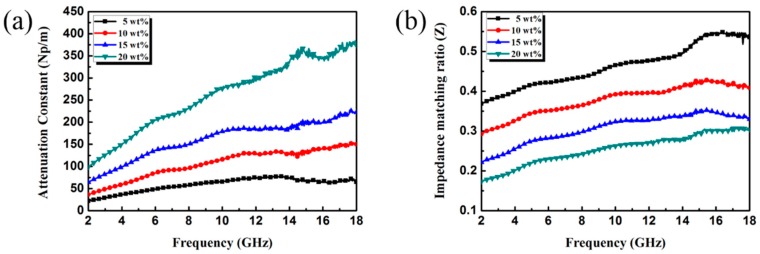
Attenuation constant (**a**) and impedance matching ratio (**b**) for paraffin composites containing different loading ratio of rGO/CuS/PPy.

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
