# Peer review of "Facile Synthesis of Sandwich-Like rGO/CuS/Polypyrrole Nanoarchitectures for Efficient Electromagnetic Absorption"

_materials, 2020, doi:10.3390/ma13020446_

Round 1

Reviewer 1 Report

In this manuscript, the authors present the synthesis and characterization of a ternary system composed by reduced graphene oxide/CuS/polypyrrole to be used for electromagnetic absorption. The subject of the research is topical and absolutely deserves promotion and publication. The main problem of the manuscript is the English language that needs for a strong improvement, maybe with the help of a native or proofreading service. The experimental needs more information on the used parameters (see attached .pdf) and figures from 5 to 8 must be enlarged because it is very difficult to interpret them, in addition all the units must be present, also if arbitrary. Other mistakes and typos are highlighted in the attached .pdf. According with these considerations the manuscript need major revision before acceptance.

Author Response

Response to reviewer 1

Dear Reviewer:

Thank you for your comments concerning our manuscript entitled “Facile synthesis of sandwich-like reduced graphene oxide/CuS/polypyrrole nanoarchitectures for efficient electromagnetic absorption” (materials-685279). These comments are all valuable and very helpful for revising and improving our paper, as well as the important guiding significance to our researches. We have studied comments carefully and have made correction which we hope meet with approval. Revised portion are marked in different color which aimed to each question in the paper. The main corrections in the paper and the response to the reviewers’ comments are as following:

Referee 1

In this manuscript, the authors present the synthesis and characterization of a ternary system composed by reduced graphene oxide/CuS/polypyrrole to be used for electromagnetic absorption. The subject of the research is topical and absolutely deserves promotion and publication. The main problem of the manuscript is the English language that needs for a strong improvement, maybe with the help of a native or proofreading service. The experimental needs more information on the used parameters (see attached .pdf) and figures from 5 to 8 must be enlarged because it is very difficult to interpret them, in addition all the units must be present, also if arbitrary. Other mistakes and typos are highlighted in the attached .pdf. According with these considerations the manuscript need major revision before acceptance.

Response: We did a thorough English proof reading on our manuscript and found a certain number of phrasing error. Some phrasing error and rewritten statements are listed below.

line 12: "How to balance the relationship of electromagnetic applications and electromagnetic pollution management is an urgent task for our human to guarantee a healthy" has been replaced by "Nowadays, electromagnetic pollution management has been highly concerned by government and society, due to the various harmful effects on human beings and wildlife " line 14: "absorbers" suggested to be "absorber", "materials" suggested to be "material" line 17: "mismatch" suggested to be "mismatches", "result" suggested to be "results" line 19: suggest to delete "facile" line 21: "composites" suggested to be "composite" line 25: suggest to delete "in this work" line 33: "against" suggested to be "for" line 33: "the" suggested to be "this" line 44: "make" suggested to be "makes" line 46: "ferromagnetic" suggested to be "Ferromagnetic" line 68: suggest to delete "which exhibit promising EA performance" line 71: suggest to add "the frequency of" line 80: suggest to delete "(Phys. Chem. Chem. Phys., 2017, 19, 14596)" line 81: suggest to delete "facile" line 82: suggest to add "the" line 86: suggest to delete " The nanoarchitecture with the filler loading of 10wt% showed a minimum RL of -49.11 dB, and effective absorption bandwidth of 4.88 GHz can be reached with the thickness of 2.5mm." line 98: suggest to delete "facile" line 112: suggest to delete "Raman spectrum", "were" suggested to be "was" line 114: suggest to add "D8-Advance, Bruker, Germany" line 115: "diffraction" suggested to be "diffractometer", suggest to add "using a Cu Kα radiation (λ = 0.15418 nm) in the scattering range (2θ) of 10-80ºat an accelerating voltage of 40 kV" line 141: suggest to add "(a)" line 144: "was" suggested to be "can be", " observed " suggested to be "appeared" line 155: "Figure 2" suggested to be "Figure 1(b)" line 180: "3.3" suggested to be "3.2" line 189: "approximately" suggested to be "close to" line 194: "change" suggested to be "changing" line 208: suggest to add "GHz" line 235: suggest to delete "both" line 250: suggest to delete "is"

Besides, the complex permittivity and complex permeability belong to the infinitesimal number and have no unit. Figures from 5 to 8 have been enlarged in the revised manuscript, and now we think the figures in the revised manuscript is clear and legible.

The changes have been marked with “    ” in the current manuscript.

.

We appreciated for your warm work earnestly, and hope that the correction will meet with approval. Once again, thank you very much for your comments and suggestions.

Reviewer 2 Report

This manuscript deals with the synthesis of new electromagnetic absorbers with significant practical applications. In the introduction the authors justify their interest in the reported work and include 37 relevant references that help the interested readers to fully understand the basics of the subject and the approach selected by the authors. In the experimental procedure, the materials required, the synthesis of sandwich-like rGO/CuS/PPy nanostructures and the characterization techniques used, namely Raman spectroscopy, FTIR, XRD,SEM, TEM and EDX are brifly described. The section on results and discussion are again relatively brief, but clear and scientifically sound for understanding their characterization of the samples and the electromagnetic absorption properties. Curiously section 3.2 does not exist!!   In summary, this is a novel work deserving publication, but the introduction can be improved, and the experimental procedures need to be further explained , particularly as far as the transmission line theory is applied. Moreover, also the English, the significance of the paper content and the quality of the paper should be improved before the paper is accepted for publication. 

Author Response

Response to reviewer 2

Dear Reviewer:

Thank you for your comments concerning our manuscript entitled “Facile synthesis of sandwich-like reduced graphene oxide/CuS/polypyrrole nanoarchitectures for efficient electromagnetic absorption” (materials-685279). These comments are all valuable and very helpful for revising and improving our paper, as well as the important guiding significance to our researches. We have studied comments carefully and have made correction which we hope meet with approval. Revised portion are marked in different color which aimed to each question in the paper. The main corrections in the paper and the response to the reviewers’ comments are as following:

Referee 2

This manuscript deals with the synthesis of new electromagnetic absorbers with significant practical applications. In the introduction the authors justify their interest in the reported work and include 37 relevant references that help the interested readers to fully understand the basics of the subject and the approach selected by the authors. In the experimental procedure, the materials required, the synthesis of sandwich-like rGO/CuS/PPy nanostructures and the characterization techniques used, namely Raman spectroscopy, FTIR, XRD,SEM, TEM and EDX are brifly described. The section on results and discussion are again relatively brief, but clear and scientifically sound for understanding their characterization of the samples and the electromagnetic absorption properties. Curiously section 3.2 does not exist!! In summary, this is a novel work deserving publication, but the introduction can be improved, and the experimental procedures need to be further explained, particularly as far as the transmission line theory is applied. Moreover, also the English, the significance of the paper content and the quality of the paper should be improved before the paper is accepted for publication.

Response: We did a thorough English proof reading on our manuscript and found a certain number of phrasing error. Section 3.3 has been corrected as 3.2 (line 180). Some phrasing error and rewritten statements are listed below.

line 12: "How to balance the relationship of electromagnetic applications and electromagnetic pollution management is an urgent task for our human to guarantee a healthy" has been replaced by "Nowadays, electromagnetic pollution management has been highly concerned by government and society, due to the various harmful effects on human beings and wildlife " line 14: "absorbers" suggested to be "absorber", "materials" suggested to be "material" line 17: "mismatch" suggested to be "mismatches", "result" suggested to be "results" line 19: suggest to delete "facile" line 21: "composites" suggested to be "composite" line 25: suggest to delete "in this work" line 33: "against" suggested to be "for" line 33: "the" suggested to be "this" line 44: "make" suggested to be "makes" line 46: "ferromagnetic" suggested to be "Ferromagnetic" line 68: suggest to delete "which exhibit promising EA performance" line 71: suggest to add "the frequency of" line 80: suggest to delete "(Phys. Chem. Chem. Phys., 2017, 19, 14596)" line 81: suggest to delete "facile" line 82: suggest to add "the" line 86: suggest to delete " The nanoarchitecture with the filler loading of 10wt% showed a minimum RL of -49.11 dB, and effective absorption bandwidth of 4.88 GHz can be reached with the thickness of 2.5mm." line 98: suggest to delete "facile" line 112: suggest to delete "Raman spectrum", "were" suggested to be "was" line 114: suggest to add "D8-Advance, Bruker, Germany" line 115: "diffraction" suggested to be "diffractometer", suggest to add "using a Cu Kα radiation (λ = 0.15418 nm) in the scattering range (2θ) of 10-80ºat an accelerating voltage of 40 kV" line 141: suggest to add "(a)" line 144: "was" suggested to be "can be", " observed " suggested to be "appeared" line 155: "Figure 2" suggested to be "Figure 1(b)" line 180: "3.3" suggested to be "3.2" line 189: "approximately" suggested to be "close to" line 194: "change" suggested to be "changing" line 208: suggest to add "GHz" line 235: suggest to delete "both" line 250: suggest to delete "is"

Besides, transmission line theory is applied to calculate the theoretical reflection loss. (line 127). The changes have been marked with Yellow tag in the current manuscript.

We appreciated for your warm work earnestly, and hope that the correction will meet with approval. Once again, thank you very much for your comments and suggestions.

Round 2

Reviewer 1 Report

The authors revised the manuscript according the suggestions and the manuscript, can be now accepted for publication.

Reviewer 2 Report

The authors considered properly the required modifications and it seems to me that the paper can be accepted for publication.